# Iterative Reconstruction of Micro Computed Tomography Scans Using Multiple Heterogeneous GPUs

**DOI:** 10.3390/s24061947

**Published:** 2024-03-18

**Authors:** Wen-Hsiang Chou, Cheng-Han Wu, Shih-Chun Jin, Jyh-Cheng Chen

**Affiliations:** 1Department of Biomedical Imaging and Radiological Sciences, National Yang Ming Chiao Tung University, Taipei 112304, Taiwan; ericwh.y@nycu.edu.tw (W.-H.C.); chw886@gm.ym.edu.tw (C.-H.W.);; 2Department of Psychiatry, Taichung Veterans General Hospital, Taichung 407219, Taiwan; 3The Human Brain Research Center, Taichung Veterans General Hospital, Taichung 407219, Taiwan; 4Department of Electro-Optical Engineering, National Taipei University of Technology, Taipei 106344, Taiwan; 5School of Medical Imaging, Xuzhou Medical University, Xuzhou 221004, China; 6Department of Biomedical Imaging and Radiological Science, China Medical University, Taichung 404333, Taiwan

**Keywords:** CT, reconstruction, iterative, OSEM, parallelism, multiple, thread, heterogeneous, GPU, finite state automaton (FSA)

## Abstract

Graphics processing units (GPUs) facilitate massive parallelism and high-capacity storage, and thus are suitable for the iterative reconstruction of ultrahigh-resolution micro computed tomography (CT) scans by on-the-fly system matrix (OTFSM) calculation using ordered subsets expectation maximization (OSEM). We propose a finite state automaton (FSA) method that facilitates iterative reconstruction using a heterogeneous multi-GPU platform through parallelizing the matrix calculations derived from a ray tracing system of ordered subsets. The FSAs perform flow control for parallel threading of the heterogeneous GPUs, which minimizes the latency of launching ordered-subsets tasks, reduces the data transfer between the main system memory and local GPU memory, and solves the memory-bound of a single GPU. In the experiments, we compared the operation efficiency of OS-MLTR for three reconstruction environments. The heterogeneous multiple GPUs with job queues for high throughput calculation speed is up to five times faster than the single GPU environment, and that speed up is nine times faster than the heterogeneous multiple GPUs with the FIFO queues of the device scheduling control. Eventually, we proposed an event-triggered FSA method for iterative reconstruction using multiple heterogeneous GPUs that solves the memory-bound issue of a single GPU at ultrahigh resolutions, and the routines of the proposed method were successfully executed on each GPU simultaneously.

## 1. Introduction

Image quality and computational efficiency are essential for analytical and iterative computed tomography (CT) reconstruction. In addition, iterative reconstruction algorithms can be separated into algebraic and statistical types. Algebraic type consists of a conventional algebraic reconstruction technique (ART), simultaneous algebraic reconstruction technique (SART), and simultaneous iterative reconstruction technique (SIRT) [1,2,3,4], while statistical type consists of maximum likelihood expectation maximization (MLEM) [5], ordered subsets expectation maximization (OSEM) [6], and maximum a posteriori expectation maximization (MAP-EM) [7]. Most of the statistical type algorithms are commonly used in nuclear medicine imaging, since they provide higher image quality than analytical methods. However, they are seldom used for CT image reconstruction because of the longer reconstruction time for a larger image matrix. Hence, general CT uses the analytical methods to reconstruct 2D (filtered back projection, FBP), and 3D images (Feldkamp–Davis–Kress, FDK) [8].

In CT image reconstruction, the projection and back-projection operations are often a fundamental performance bottleneck, especially for micro-computed tomography (micro-CT) scans with ultrahigh resolution. In the last decade, advances in semiconductors have led to the development of high-performance computing techniques that use graphics processing units (GPUs) to massively improve the efficiency of CT image reconstruction. GPU-based CT reconstruction has obtained spectacular acceleration factors, and some manufacturers now rely on this technology for their clinical systems [9,10,11,12,13,14,15,16]. For example, Hofmann et al. [17] compared CPU and GPU implementations of the FDK algorithm with different combinations of hardware and application programming interfaces (APIs) and found that the latter realized an acceleration of 200 times compared with the former.

Iterative reconstruction methods are more complex than analytical reconstruction methods to adapt for GPU implementation, and new strategies must be developed to achieve significant acceleration factors. One approach has been to use multiple GPUs simultaneously to perform the projection and back-projection operations as well as the system matrix calculations of the MLEM algorithm. For high-resolution CT image reconstruction, the entire full-scan view system matrix is difficult to store in the dynamic random access memory (DRAM) of a computer [18]. On-the-fly system matrix (OTFSM) calculation using a GPU can reduce the data transfer time between its local memory and the primary DRAM of the computer, which reduces the reconstruction time compared with using a pre-calculated system matrix stored in the primary DRAM. However, Nvidia proposed a peer-to-peer CUDA link to increase data transfer bandwidth between GPUs and host main memory; NVLink/NVSwitch provided 900 GB/s, nearly seven times more than the PCIe Gen 5. Also, it unifies and manages the local memory of GPUs using the NVLink/NVSwitch. It is supported in high-performance server systems (P100, V100, A100 and H100). Unfortunately, Nvidia’s low-cost GPUs do not support this technique. Hence, memory unified management between GPUs and tasks operation assignment to the GPU were required for consideration.

Multi-GPU image reconstruction has been widely discussed in several articles [19,20,21,22,23]. The difficulty of scaling applications for reconstruction algorithms on multi-GPU configurations is often underestimated. Task-pipelined simultaneous concurrence execution on multi-GPU is a crucial feature for the acceleration of iterative reconstruction time. However, the selection of a suitable parallel reconstruction algorithm is essential. It needs to separate the independent operation task for the iterative algorithm, which can simultaneously execute on multi-GPUs. This key feature will dominate the run time performance for the iterative reconstruction algorithm. Homogeneous multi-GPU reconstruction has several methods for speedup, using the specific memory of GPU (texture or shared memory), task pipeline for multi-GPUs (assign forward-projection, back-projection, filtering tasks for dedicated GPU, and pipelining operation for reconstruction), and slice-based image reconstruction for each GPU. Homogeneous multi-GPU image reconstruction is hard to work on the cloud system because it lacks a handshaking and task flow control mechanism between the cloud nodes. We propose an event-triggered finite state automaton (FSA) method for iterative reconstruction of ultrahigh-resolution micro-CT scans using multiple heterogeneous GPUS performing OTFSM calculation for OSEM. The proposed method was developed to mitigate the factors that affect statistical iterative reconstruction, obtain high-quality images from low-dose partial scan views, and reduce the reconstruction time, data transfer overhead, and dependency on a high-capacity DRAM. 

The motivation for our proposed method, driven by the need to manage communication and synchronization between heterogeneous GPUs effectively, serves as a crucial stepping stone toward tackling the challenges of managing large-scale, data-intensive computations in cloud environments. The operational behavior of GPUs closely resembles that of numerous cloud nodes, motivating us to investigate their potential in this context. Our study leveraged multiple heterogeneous GPUs on a workstation to demonstrate efficient management of concurrent computations, paving the way for future cloud-based applications. However, managing multiple heterogeneous GPUs concurrently across cloud nodes presents a significant challenge due to complex data synchronization issues arising from varying data transfer latencies among master and slave nodes. By leveraging the similarity between managing GPUs and cloud nodes, our method can be readily implemented on any cloud system, maximizing resource utilization and enhancing the efficiency of iterative reconstruction within the network.

## 2. Background Information

For a single GPU, OTFSM calculation for OSEM is a maximum likelihood for transmission tomography (MLTR) algorithm, which is derived from the Beer–Lambert law, maximum log-likelihood estimation [24,25,26], and a refinement step. These are, respectively presented below:(1)$$\hat{y_{i}}=b_{i}\cdot \exp\left(-\overset{J}{\underset{j=1}{\sum }}l_{ij}\cdot \mu_{j}\right)$$
(2)$$L=\overset{I}{\underset{i=1}{\sum }}(y_{i}\cdot \ln\left({\hat{y}}_{i}\right)-{\hat{y}}_{i})$$
(3)$$\Delta \mu_{j}=-\frac{\frac{\partial L}{\partial \mu_{j}}\left(\vec{\mu }\right)}{\sum_{h=1}^{j}\frac{\partial^{2}L}{\partial \mu_{j}\partial \mu_{h}}\left(\vec{\mu }\right)},$$
where $\hat{y_{i}}$ is the *i*-th pixel value of an estimated projection, $y_{i}$ is the *i*-th pixel value of a measured projection, $\mu_{j}$ is the image pixel value of the *j*-th ray passing through the image into *i*-th pixel value of an estimated projection. $b_{i}$ is the *i*-th pixel value of the projection acquired from the blank scan. $l_{ij}$ is the weight of the *j*-th ray passing through the *i*-th pixel of the system matrix, and $l_{ih}$ is a regularization term that accumulates $l_{ij}$ for each scanning degree. Substituting Equation (2) into Equation (3) [27,28,29], we obtain the MLTR algorithm:(4)$$\mu_{j}^{n+1}=\mu_{j}^{n}+\frac{\sum_{i=1}^{I}l_{ij}\cdot \left(\hat{y_{i}}-y_{i}\right)}{\sum_{i=1}^{I}l_{ij}\cdot \left[\sum_{h=1}^{J}l_{ih}\right]\cdot \hat{y_{i}}}$$
where *n* is the iteration number. Fessler et al. [30] derived a class of algorithms that monotonically increases the likelihood in Equation (4), which can be used in OTFSM calculation for ordered subsets MLTR (OS-MLTR) via a single GPU, as described in Pseudocode in Algorithm 1. A ray-tracing technique [31,32] can be used to calculate the pixel weights of the system matrix.
**Algorithm 1:** OTFSM calculation for OS-MLTR using a single GPUInput: Projection, AngleList, StartSlice, EndSlice, BlankProj, TotalProjCnt, SetNumber, SubSetNumber, GPU_ID, SetIterCnt;Output: SliceImage SetCudaDevice(GPU_ID);/Activates a CUDA device with the GPU_ID***float*** ts = SumSM(0→TotalProjCnt);//The term of [∑h=1Jlih]   in Equation (4)***float*** cur_proj_angle;***uint***      SetIndex, SliceN, cur_proj_index; boolean StopCriteria;***float*** SM, SMT;      //Dynamic allocation and transpose of the system matrix***float*** U1, U2, cur_guess, y, y^, P;***for*** SliceN: StartSlice→EndSlice         ***While***(! StopCriteria || iter_cnt < SetIterCnt){                           ***for*** SetIndex: 0→SetNumber         //OS-MLTR outer loop                                    SetIndex = 0; U1 = 0; U2 = 0; cur_guess = 0;                                    ***for*** SubSetIndex: 0→SubSetNumber         //OS-MLTR inner loop                                             cur_proj_index = SubSetIndex × SetNumber + SetIndex;                                             cur_proj_angle = AngleList[cur_proj_index];                                             y = Projection[cur_proj_index];                                             gpu_OTFSM(cur_proj_angle, &SM, &SMT);                                             P = SMT × cur_guess;         //Forward-projection                                             y^   = BlankProj. × exp(−P);                                             U1 = SM × ( y^−y) + U1;         //Backprojection                                             U2 = SM × (ts.∗y^) + U2;                                    cur_guess = (U1./U2) + cur_guess;         SliceImage = cur_guess;          StopCriteria = Check_StoppingCriteria();//Stopping Criteria is RMSE

## 3. Materials and Methods

### 3.1. Proposed Method

The proposed method uses heterogeneous GPUs to perform parallel ray-tracing weight calculations within a subset of the system matrix to improve the memory efficiency, input/output access, and program execution capabilities of a single GPU. This paper introduces two distinct event-triggered FSAs for heterogeneous multi-GPU control within the OS-MLTR. The first leverages a simple device FIFO queue for concurrent GPU tasks, while the second utilizes a job queue to achieve higher system matrix calculation throughput and minimize coordinated time through threaded variable synchronization. Pseudocode in Algorithm 2 shows the parallel processing and coordinated parallelism task operations for OS-MLTR using heterogeneous GPUs and includes three FSAs, as shown in Figure 1: OSEM_M, OSEM_Inner_S1, and OSEM_Inner_S2. The function Launch_FSA() triggers OSEM_M, OSEM_Inner_S1, and OSEM_Inner_S2, which are waiting for an event to perform the parallelism tasks of the GPUs. The function Gather_available_GPU() gathers the available idle GPUs in the system.
**Algorithm 2:** FSAs for OTFSM calculation for OS-MLTR using heterogeneous GPUsInput: Projection, AngleList, StartSlice, EndSlice, BlankProj, TotalProjCnt, SetNumber, SubSetNumber, SetIterCnt;Output: SliceImage***struct*** recon_param_struct;***uint*** SetIndex, SliceN, TotalGPU, StopCriteria;TotalGPU = Gather_available_GPU(); ***float*** ts = SumSM(0→TotalProjCnt);//The term of [∑h=1Jlih]   in Equation (4)***uint*** gpu_Idle_queue[TotalGPU], gpu_suspend_queue[TotalGPU];//The flow control queue***pthread_t*** pthd_idle[TotalGPU], pthd_suspend[TotalGPU];***float*** U1[TotalGPU], U2[TotalGPU], cur_guess;***for*** SliceN: StartSlice→EndSlice      ***While***(! StopCriteria || iter_cnt < SetIterCnt)            **for** SetIndex: 0→SetNumber            SetIndex = 0; cur_guess = 0; U1[0→TotalGPU] = 0; U2[0→TotalGPU] = 0;            **Launch_FSA**(OSEM_M, OSEM_Inner_S1, OSEM_Inner_s2);            ***While*** (SetIndex == SetNumber)                  ***if***(!is_empty(gpu_idle_queue)& is_get_subset_index)                        recon_param_struct->proj_index=SubSetIndex×SetNumber+SetIndex;                        gpu_id = pop(gpu_idle_queu); SetIndex++;                         ***if***(OSEM_Inner_s1.S7)                         pthd_idle[gpu_id] = pthread_create(func_inner_s1, gpu_id);                        ***if***(!is_empty(gpu_suspend_queue)& is_ready(cur_guess))                        inner_s2_id = pop(gpu_suspend_queue);                        pthd_suspedn[gpu_id]=pthread_create(func_inner_s2,inner_s2_id);                        ***if***(check_pthd_join(pthd_idle[indx:0→TotalGPU]))                        gpu_id = indx; push(gpu_idle_queue, gpu_id);                        ***if***(check_pthd_join(pthd_suspend[inner_s2_id])                        push(gpu_idle_queue, inner_s2_id);                  ***if***(is_OSEM_M.S5 & !is_empty(gpu_idle_queue)                  gpu_id = pop(gpu_idle_queu);                  pthd_idle[gpu_id] = pthread_create(func_outer, gpu_id);                  ***if***(check_pthd_join(pthd_idle[gpu_id]))            gpu_id = indx; push(gpu_idle_queue, gpu_id);       SliceImage = cur_guess;       StopCriteria = Check_StoppingCriteria();//Stopping Criteria is RMSE

Pseudocodes in Algorithms 3 and 4 show the two parallel-step operations of the ordered subsets: OTFSM calculation and refinement of the back-projection calculation of the current predicted image. The function *pthread_creat()* starts a new thread in the calling process of the operating system. The new thread is executed by invoking the routines *func_inner_s1*, *func_inner_s2*, or *func_outer*. *func_inner_s1* performs the OTFSM calculation. *func_inner_s2* refines the current back-projection image. *func_outer* calculates the current predicted image based on the results of *func_inner_s2*.
**Algorithm 3:** OTFSM calculationFunction: func_inner_s1(gpu_id, reco_param_struc){setCudaDevice(gpu_id);***uint***  proj_index = reco_param_struct->proj_index;***float***  SM = recon_param_struc->SM;***float***  SMT = recon_param_struct->SMT;***float***  proj_angle = recon_param_struct->AngleList[proj_index];**gpu_OTFSM**(proj_angle, SM, SMT);}

**Algorithm 4:** Refinement step for back-projection calculationFunction: func_inner_s2(gpu_id, cur_guess, recon_param_struct){setCudaDevice(gpu_id);***uint*** proj_index = reco_param_struct->proj_index;***float***   SM = recon_param_struc->SM;***float***   SMT = recon_param_struct->SMT;***float***   U1 = recon_param_struct->U1;***float***   U2 = recon_param_struct->U2;***float***   y = Projection[proj_index];***float***   P = SMT × cur_guess;***float***   y^= BlankProj. × exp(−P);U1 = SM × ( y^−y) + U1;U2 = SM × (ts. × y^) + U2;}

OS-MLTR comprises two loop blocks, where the outer loop controls which subsets run into the inner loop. The outer loop event trigger flow control through the FSA OSEM_M. The inner loop includes two FSAs that control the flow: OSEM_inner_S1, and OSEM_inner_S2. Table 1 defines the states of OSEM_M. State S0 is the initialization task, which pushes all available idle GPUs into the idle queue and causes OSEM_M to transition to State S1. It also empties the suspended queue, so suspended GPUs wait for the current predicted image to be released by the locked GPU before performing back-projection and handshaking for initialization. In State S2, the idle queue is monitored to determine whether it is empty. If the idle queue is empty, OSEM_M remains in State S2. If the idle queue is not empty, OSEM_M receives a subset index from the ordered subsets. Then, an idle GPU is taken from the idle queue, and OSEM_M transitions from State S2 to State S3. OSEM_M creates a parallel thread to process a subroutine and transfer flow control authority to the OSEM inner loop, and it transitions to State S4. When the OSEM inner loop routine has been completed, flow control transfers back to OSEM_M, which transitions back to State S2. In State S5, OSEM_M creates a thread to process a subroutine of the outer loop. Pseudocode in Algorithm 5 calculates the current predicted image from the refinement step. Table 2 defines the conditions for state transitions of OSEM_M.
**Algorithm 5:** Calculation of the current predicted imageFunction: func_outer(gpu_id, cur_guess, recon_param_struct){setCudaDevice(gpu_id);***float*** U1 = recon_param_struct->U1;***float*** U2 = recon_param_struct->U2;cur_guess = (U1./U2) + cur_guess;}

Table 3 defines the states for OSEM_Inner_S1. In State S0, OSEM_Inner_S1 initializes the image scanning and reconstruction parameters, sets the local memory usage threshold for the GPUs, gathers the available GPUs, and creates a lookup table for the sin and cos values used in ray tracing for weighted OTFSM calculation. In State S1, the idle queue is empty. In State S2, a subset index for the angle of the scanning object center is received. In State S3, the idle queue is not empty, and a candidate *gpu_id* is drawn from the queue. In State S4, OSEM_Inner_S1 calculates the allocated memory size of the system matrix with the subset index from State S2. In State S5, OSEM_Inner_S1 checks whether the local memory usage of the candidate GPU is above or below the threshold. In State S6, the local memory usage is above the threshold, and the candidate *gpu_id* is pushed into the suspended queue. OSEM_Inner_S1 waits for the current predicted image to be released by the locked GPU, and it returns to State S0. State S7 is similar to State S3 of OSEM_M, and OSEM_Inner_S1 creates a thread to process a subroutine of Pseudocode in Algorithm 2. In State S8, OSEM_Inner_S1 calculates all angles of the subset and sets the semaphores for the current subset. Table 4 describes the conditions for the state transitions of OSEM_Inner_S1.

Table 5 defines the states of OSEM_Inner_S2. In State S1, OSEM_Inner_S2 is waiting for the current predicted image to be ready. In State S2, it draws *gpu_id* from the suspended queue when it is not empty. In State S3, it creates a thread to process a subroutine of Pseudocode in Algorithm 2, which processes the refined value. In State S4, it is waiting for the thread to join the main process. In State S5, it pushes *gpu_id* to the idle queue and then returns to the idle State S0. GPUs in the suspended queue are determined to exceed the upper bound for memory usage by OSEM_Inner_S1. OSEM_Inner_S2 performs Pseudocode in Algorithm 2 for iterative calculation of each projection. Table 6 describes the conditions for the state transitions of OSEM_Inner_S2. The proposed event-trigger FSA flow control mechanism has two states (GET_IDLE_DEVICE: OSEM_Inner_S1.S3 and GET_SUSPEND_DEVICE: OSEM_Inner_S2.S2), and select the identified number of candidate GPU, which was popped up from the idle device queue or suspended device queue. The Nvidia CUDA API function of “cudaSetDevice(gpu_id)” sets the CUDA kernel task to run in the dedicated GPU. 

The proposed method within the second framework significantly enhances the throughput of system matrix calculations on GPUs and concurrently diminishes the synchronization time associated with controlling these calculations. Table 7, Table 8, Table 9 and Table 10 comprehensively detail the state definitions and transition conditions guiding this behavior. To achieve this improvement, we implemented a mutual exclusion mechanism utilizing the portable operating system interface of the application programming interface (POSIX API) library for access control within critical threading regions. Specifically, the mutual exclusion functions ensure exclusive access and maintain efficient synchronization. It is important to note that the main state machine structure remains comparable to the version employing the FIFO device queue. The main state machine is the same as the FIFO device queue version. But we used the disable flag to bypass the OSEM_M.S2 (CHECK_DEVICE_QUEUE) and disable the idle and suspended queue. Meanwhile, within OSEM_Inner_S1.S2 (SET_JOB_Q state), the subset index associated with the system matrix calculation is directly inserted into the GPU job queue. This eliminates the overhead of waiting for the device queue to become available, further streamlining the process. In this state, the GPUs remain idle, awaiting the completion of the refinement operation. To coordinate the concurrent execution of system matrix calculations and refinement operations on the GPUs, we employ several thread synchronization variables. These variables, defined in the state transition tables in Table 9 and Table 10, facilitate effective communication and collaboration among the processing units.

### 3.2. Experiments

The proposed method was implemented using CUDA C on a heterogeneous multi-GPU system. The CUDA driver version is 10.1. Each GPU has two copy engines for concurrent copy and kernel execution. The system comprised an Intel Core I7-9800x CPU (3.8 GHz, 8 cores, 16 threads, 32 GB main memory; Intel: Santa Clara, CA, USA) and three GPUs, whose specifications are listed in Table 11: NVIDIA TITAN_Xp, NVIDIA GeForce GTX1060, and NVIDIA GeForce GTX1050 (NVIDIA: Santa Clara, CA, USA). OSEM_Inner_S1 was used to calculate the pixel weights of the system matrix through ray tracing. 

Experimental data were collected by using a custom-built micro-CT in our laboratory. CT scanning was performed under dose conditions with a tube voltage of 70 kV, current of 0.05 mAs (0.838 µGy/s). The image detector had a pixel size of 0.075 mm and matrix size of 1944 × 1536. The experimental phantom was QRM HA phantom, which houses five cylindrical inserts containing various densities of calcium hydroxyapatite (HA0, HA100, HA200, HA400, and HA800). HA densities specified 0, 100, 200, 400 and 800 mg HA/${\mathrm{cm}}^{3}$. All the images were reconstructed using OS-MLTR algorithms with similar spatial resolution. The subset of OS-MLTR was 10, and the iteration number was 10 (stopping criterion: root mean square error < 0.015). The root mean squared error (RMSE) was used as the stopping criterion for Pseudocode in Algorithm 2. 

## 4. Results

Figure 2A,C shows the target object of the CT scanning, which is a QRM HA phantom. Figure 2B shows reconstructed image produced using OS-MLTR using our proposed multiple GPUs. (SID = 127 mm, SOD = 78.5928 mm).

Table 12 compares the runtimes of three scenarios using OS-MLTR image reconstruction. Scenario A involves image reconstruction on a single GPU of Nvidia Titan-XP. Scenario B utilizes a first-in-first-out (FIFO) device scheduling algorithm for heterogeneous multiple GPUs. Scenario D has the same hardware settings as Scenario B. However, they employ different device task scheduling methods, utilizing the job queue to increase the speed of system matrix calculations on the GPU. Scenario C consists of two identical GPUs, one of Titan and one of the GTX-1050 and uses the job queue method to reconstruct image. We also compared homogeneous and heterogeneous scenarios using job queue scheduling. Scenarios E, F, G, H, and I involve combinations of two GPUs. Homogeneous scenarios consist of two identical GPUs of either Titan or GTX-1050, while heterogeneous scenarios involve combinations of Titan, GTX-1060, and GTX-1050, respectively. 

Figure 3 shows the reconstructed images produced using OS-MLTR using our proposed multiple GPUs methods. Three scenarios were considered: (s1) all scan views (reconstructed from 600 projections), (s5) one-fifth of all scan views (reconstructed from 120 projections), and (s10) one-tenth of all scan views (reconstructed from 60 projections). These scenarios aim to evaluate the impact of using fewer projections on image quality and runtime at different sampling levels. Table 13 shows the SNR and runtime evaluation results, providing a runtime selection that balances the accepted image quality tradeoff.

## 5. Discussion

In the experimental results of Table 12, Scenario B demonstrated a drawback in that the image reconstruction performance with multiple GPUs was lower than with a single GPU (i.e., NVIDIA Titan-XP). This can be attributed to the lower computation capacity of GTX-1050, which blocked GPUs with a higher computation capacity in the idle or suspended queues. In other words, GPUs with a high computation capacity (i.e., Titan-XP or GTX-1060) had to wait for GTX-1050 to obtain the projection index for OTFSM calculation or to obtain the predicted image for the refinement step. The idle and suspended queues followed a first-in-first-out (FIFO) scheduling strategy, which degraded the computational efficiency when the lowest-performing GPU was at the head of the queue and increased the idle time of the highest-performing GPUs. Another factor impacting runtime performance is the synchronization of thread task statuses. Our current implementation checks the event trigger status only after the response from the pthread_join POSIX API function. This method lacks real-time synchronization of the FSA state during thread execution on the GPUs. To address this issue, we propose a device scheduling mechanism that improves the throughput of system matrix calculations for heterogeneous multi-GPUs and reduces the synchronization time for thread task execution on the GPUs. Specifically, we enhance the throughput of system matrix calculations by adding a job queue. Furthermore, to ensure synchronized access to critical regions within the threading process, we implemented a mutual exclusion (mutex) mechanism provided by the POSIX API library. This mechanism leverages the functions pthread_mutex_lock and pthread_mutex_unlock to establish exclusive access during critical operations, thereby preventing race conditions and guaranteeing data integrity. Within the OSEM_Inner_S1.S2 (SET_JOB_Q state), the subset index associated with the system matrix calculation is directly inserted into the GPU job queue. Our experimental setup utilizes three heterogeneous GPUs. We manually assign specific payloads to the job queue using this format: JOB_Q(n1, n2, n3). n1 was the elements of the degree index associated with the subset for GPU1, n2 was for GPU2, and n3 for GPU3, respectively. The values of n1, n2, and n3 are determined based on the individual memory capacities of each GPU. The OSEM_Inner_S1.S5 (SUBSET_PENDNG) state holds subsets of the system matrix calculation that cannot be processed in a single run. These subsets remain in this state until the refinement operation is completed. To coordinate the concurrent execution of system matrix calculations and refinement operations across the GPUs, we introduce several thread synchronization variables. These variables are defined in the state transition tables provided in Table 10 and Table 11. Table 12 presents the best-run performance of image reconstruction achieved using our proposed method featuring a high-throughput job queue on heterogeneous multi-GPUs. The table compares Scenario A and Scenario C, demonstrating that Scenario C outperforms Scenario A with a 5× speedup. Additionally, for images of size 3888 × 3888, Scenario C exhibits a 9× speedup compared to Scenario B. These performance gains highlight the efficacy of our proposed job queue approach. The homogeneous multiple GPUs represent a specific case within our proposed heterogeneous multiple GPUs method. The corresponding reconstruction times for all scenarios were shown in Table 12. We observed an approximately linear increase in execution time across Scenarios C, D, and E for identical image reconstruction programs. While a configuration of three identical Titan-XP GPUs would likely exhibit superior performance compared to three heterogeneous GPUs, our proposed method demonstrates flexibility in adapting to diverse GPU environments, not restricted solely to homogeneous configurations. This flexibility is achieved by leveraging the CUDA API library functions cudaGetDeviceCount and cudaGetDeviceProperties. These functions allow us to determine the number of available GPUs and their individual properties, enabling us to efficiently assign tasks from the job queue (containing *n* elements) to the appropriate GPUs based on their capabilities. The results unveiled an intuitive trend in runtime, suggesting that configurations with higher computational capacity GPUs yield superior reconstructed performance. We evaluated the runtime performance across scenarios involving two GPUs and three GPUs, specifically comparing Scenarios C and E. Both scenarios featured two identical Titan GPUs, with Scenario C including an extra GTX-1050 GPU. In theory, Scenario C should exhibit better runtime performance than Scenario E. However, the results demonstrate the opposite. Two primary factors contribute to this discrepancy. Firstly, Scenario C incurs additional overhead in the control variable for synchronization with three GPUs. Secondly, the management of event trigger flags in the state transition condition process is handled by a branch condition statement in the program. This leads to an excessive number of event trigger flags, resulting in unnecessary delays during state transitions. From the homogeneous and heterogeneous experimental results, we observed that the runtime does not decrease with an increase in the computational capacity of the GPU at small, reconstructed matrix sizes (243 × 243, 486 × 486). This is because the calculation turnaround rate of heterogeneous GPUs is nearly equal, leading to a race condition for the synchronization of the FSA control variable. However, we found that the image reconstruction performance is affected by the race condition of the control variable synchronization for matrices larger than matrix size of 486 × 486, although the impact was minor.

The image regularization term could be processed on the FSA of OSEM_Inner_s2. The state machine adds a new state after the OSEM_Inner_s2.S3. From Figure 3, we have found that all reconstructions seem to be misaligned. This may be attributed to the offset of the scanner holder. But the geometric calibration of the image is beyond the scope of this study. We also found an increase in CT-values at the bottom of the table and the top of the phantom. Since there is no any beam hardening correction algorithm applied, the cupping artifact (lower CT-value) and scatter affects the CT image and causes higher CT-value to occur.

## 6. Conclusions

We proposed an event-triggered FSA method for iterative reconstruction using multiple heterogeneous GPUs that solves the memory bound issue of a single GPU at ultrahigh resolution. Also, our method can provide a task parallelism concurrent control mechanism for the OS-MLTR algorithm. The results verified the system matrix calculation task or the estimated image’s refinement step, which were simultaneously running on the heterogeneous GPUs. Our proposed method can be expanded to a higher number of multiple heterogeneous GPUs, and even has the potential to be used in cloud computing with multiple heterogeneous GPUs. The experimental results show the method speeds up five times faster than the scenario of the single GPU and nine times faster than a heterogeneous numerous GPU with a device FIFO queue; this was improved by the job queue to increase the task operation throughput of the GPUs and adds threading synchronization variables to coordinate the task operation on GPUs. In both homogeneous and heterogeneous scenarios, our method proved to be flexible and easy to implement in an environment consisting of multiple identical or non-identical GPUs, without adding an extra burden to the system. Future work will involve implementing an auto-loading balance for task assignments in the job queue to enhance the manual fine-tuning of tasks. This process could be easily replicated on cloud nodes.

## Figures and Tables

**Figure 1 sensors-24-01947-f001:**
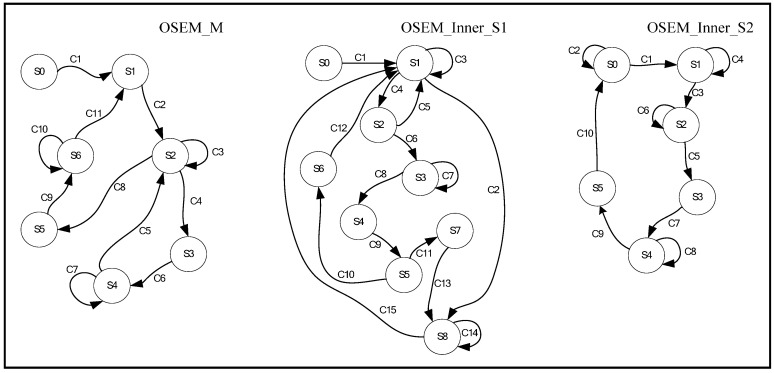
FSAs for flow control of OTFSM calculation for OS-MLTR using heterogeneous GPUs.

**Figure 2 sensors-24-01947-f002:**
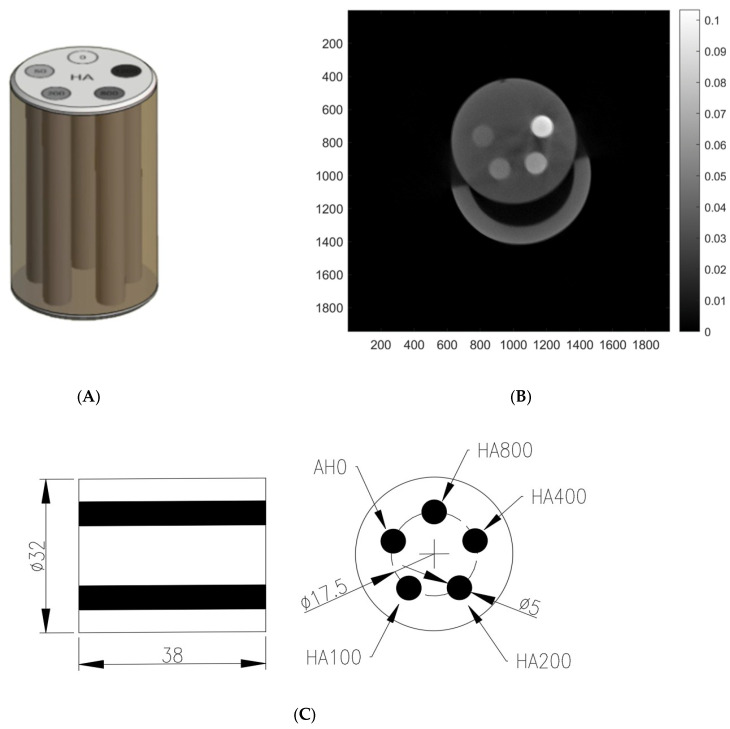
(**A**) QRM HA phantom. (**B**) Image reconstructed using OS-MLTR from a full scan condition, and (**C**) QRM HA phantom five cylindrical sectional view.

**Figure 3 sensors-24-01947-f003:**
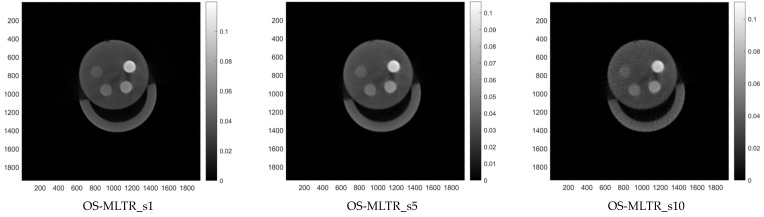
CT images reconstructed with OS-MLTR algorithm method. (SID = 127 mm, SOD = 78.59 mm, Detector size = 1944 × 1536 pixels, detector pixel size = 0.075 × 0.075 mm, reconstruction image = 1944 × 1944 pixels).

**Table 1 sensors-24-01947-t001:** State definitions of OSEM_M for FIFO of device control.

State No.	State Name
S0	INITIAL
S1	IDLE
S2	CHECK_DEVICE_QUEUE
S3	CREATE_THREAD_OF_OSEM_INNER
S4	WAIT_OSEM_INNER_COMPLETE
S5	OSEM_OUTER
S6	OSEM_OUTER_COMPLETE

**Table 2 sensors-24-01947-t002:** Conditions for state transitions of OSEM_M for FIFO of device control.

Condition	State Trans.	Description
C1	S0→S1	OS-MLTR parameters are initialized.
C2	S1→S2	Set the flow control to the initial state and build up the idle and suspended queues.
C3	S2→S2	The idle queue is empty.
C4	S2→S3	The idle queue is not empty.
C5	S4→S2	The OSEM inner loop is completed.
C6	S3→S4	The function of the OSEM inner loop is launched.
C7	S4→S4	The OSEM inner loop is not complete.
C8	S2→S5	The OSEM inner loop is complete, and the idle queue is empty.
C9	S5→S6	The function of the OSEM outer loop is launched.
C10	S6→S6	The OSEM outer loop is not complete.
C11	S6→S1	The OSEM outer loop is completed.

**Table 3 sensors-24-01947-t003:** State definitions of OSEM_Inner_S1 for FIFO of device control.

State No.	State Name
S0	OSEM_S1_INITIAL
S1	OSEM_S1_IDLE
S2	GET_SUBSET_INDEX
S3	GET_IDLE_DEVIC
S4	CHECK_DEVICE_MEM
S5	WAIT_DEVICE_MEM_CHECK
S6	SUSPEND_DEVICE_QUEUE
S7	PTH_CREATE_OSEM_S1
S8	STOP_OSEM_S1_FSM

**Table 4 sensors-24-01947-t004:** Conditions for state transitions of OSEM_Inner_S1 for FIFO of device control.

Condition	State Trans.	Description
C1	S0→S1	Set image scanning parameters, gather properties of available GPUs, and flush the idle and suspended queues.
C2	S1→S8	SM calculation of the last projection with the subset has been completed.
C3	S1→S1	Waiting for an idle GPU.
C4	S1→S2	The idle queue is not empty.
C5	S2→S1	Waiting for an idle GPU.
C6	S2→S3	Obtain a candidate GPU ID from the idle queue.
C7	S3→S3	Waiting for an idle GPU
C8	S3→S4	Check the onboard memory usage of the candidate GPU
C9	S4→S5	Waiting for the memory usage of the candidate GPU to be checked
C10	S5→S6	The memory usage is over the limit, and the current subset index is abandoned; the GPU ID is pushed into the suspended queue.
C11	S5→S7	Launch a thread task for OTFSM calculation with the candidate GPU
C12	S6→S1	Push the GPU ID into the suspended queue and wait for the current predicted image to be ready
C13	S7→S8	Return to the idle state
C14	S8→S8	Completed OTFSM calculation of the subset
C15	S8→S1	Return to the idle state

**Table 5 sensors-24-01947-t005:** State definitions of OSEM_Inner_S2 for FIFO of device control.

State No.	State Name
S0	OSEM_S2_IDLE
S1	GET_CURRENT_GUESS
S2	GET_SUSPEND_DEVICE
S3	PTH_CREATE_OSEM_S2
S4	OSEM_INNER_S2_COMPLETE
S5	SET_DEVICE_IDEL

**Table 6 sensors-24-01947-t006:** Conditions for state transitions of OSEM_Inner_S2 for FIFO of device control.

Condition	State Trans.	Description
C1	S0→S1	The suspended queue is not empty.
C2	S0→S0	The suspended queue is empty.
C3	S1→S2	The current predicted image is ready.
C4	S1→S1	The current predicted image token is busy.
C5	S2→S3	Draw a candidate GPU from the suspended queue, and assign the current predicted token.
C6	S2→S2	Wait until the suspended queue is not empty.
C7	S3→S4	Launch a thread task to update the predicted image with the candidate GPU.
C8	S4→S4	Wait for the GPU to update the newest current predicted image.
C9	S4→S5	Complete the thread task.
C10	S5→S0	Push the candidate GPU back to the idle queue and return to the idle state.

**Table 7 sensors-24-01947-t007:** The state of stage s1 for the job queue scheduling.

State No.	State Name
S0	OSEM_S1_INITIAL
S1	OSEM_S1_IDLE
S2	SET_JOB_Q
S3	PTH_CREATE_OSEM_S1
S4	WAIT_JOB_COMPLETE
S5	SUBSET_PENDING
S6	STOP_OSEM_S1_FSM

**Table 8 sensors-24-01947-t008:** The state of stage s2 for the job queue scheduling.

State No.	State Name
S0	OSEM_S2_IDLE
S1	GET_CURRENT_GUESS
S3	PTH_CREATE_OSEM_S2
S4	OSEM_INNER_S2_COMPLETE
S5	SET_DEVICE_IDEL

**Table 9 sensors-24-01947-t009:** Conditions for state transitions of job queue scheduling with OSEM_Inner_S1.

Condition	State Trans.	Description
C1	S0→S1	Set the initial value for the threading synchronization control variables
C2	S1→S2	The threading synchronization variable of the device idle is true
C3	S1→S6	The threading synchronization variable of the last projection with the subset is true
C4	S2→S3	Set the degree index in the job queue for the system matrix calculation
C5	S2→S1	The synchronization variable of the device idle is false
C6	S2→S6	The threading synchronization variable of the last projection with the subset is true
C7	S3→S4	Create a threading task to process system matrix calculation with the job queue
C8	S4→S6	The threading synchronization variable of the last projection with the subset is true
C9	S4→S5	Wait for a threading task of the system matrix calculation completion
C10	S5→S1	Set the threading synchronization variable of the device suspend that waits for refinement operation
C11	S6→S6	Wait for the refinement operation completion
C12	S6→S1	The threading synchronization variable of the last projection with the subset is true, and the refinement operation completion

**Table 10 sensors-24-01947-t010:** Conditions for state transitions of job queue scheduling with OSEM_Inner_S2.

Condition	State Trans.	Description
C1	S0→S0	Wait for the system matrix calculation completion of the job queue
C2	S0→S1	System matrix calculation completion
C3	S1→S1	Wait for current guess image is ready and to do the refinement operation
C4	S1→S2	Current guess image ready and create a thread task to do refinement operation
C5	S2→S2	The device threading synchronization variable of refinement operation completion is false
C6	S2→S3	The device threading synchronization variable of refinement operation completion is true
C7	S3→S3	Wait for the refinement operation completion
C8	S3→S4	The refinement operation completion
C9	S4→S0	Set the threading synchronization of device status to idle

**Table 11 sensors-24-01947-t011:** Specifications of the GPUs.

Nvidia GPU Type	CUDA Core	Memory Size	Memory Width	Bandwidth
Titan-Xp	3840	12 GB	384 bits	547 GB/s
GeForce GTX1060	1280	6 GB	192 bits	192 GB/s
GeForce GTX1050	768	4 GB	128 bits	112 GB/s

**Table 12 sensors-24-01947-t012:** The run time ^1^ of five image reconstruction matrix ^2, 3^ for nine scenarios ^5, 6, 7, 8^.

Scenario	GPUs ^4^	243 × 243	486 × 486	972 × 972	1944 × 1944	3888 × 3888
A ^5^	1	7.27	17.05	40.58	127.83	479.69
B ^6^	1, 2, 3	9.84	25.24	74.43	279.76	923.18
C ^7^	1, 1, 3	1.39	3.11	7.91	24.68	95.30
D ^7^	1, 2, 3	2.10	5.23	11.99	34.66	124.20
E ^8^	1, 1	1.32	2.80	7.17	21.60	81.12
F ^8^	1, 2	1.98	4.09	9.58	29.00	114.73
G ^8^	1, 3	1.76	3.89	9.99	31.30	122.52
H ^8^	2, 3	2.54	6.05	17.59	64.37	275.21
I ^8^	3, 3	2.43	6.21	19.56	76.81	333.29

^1^ Time unit: s/per-subset. ^2^ Digital phantom: 360 projections (OS-MLTR: 10 subsets, 36 projection/per-subset). ^3^ Reconstruction image matrix: 243, 486, 972, 1944 and 3888. ^4^ GPU No.: 1. Taitan-XP, 2. GTX-1060, 3. GTX-1050. ^5^ Scenario A: Single GPU for OS-MLTR (Nvidia Tatin-XP). ^6^ Scenario B: heterogeneous multi-GPU with FIFO device scheduling for OS-MLTR.(3 GPUs). ^7^ Scenario C, D: heterogeneous multi-GPU with job queue for OSM-MLTR. (3 GPUs) ^8^ Scenario E, F, G, H, and I: homogeneous or heterogeneous multi-GPU with job queue for OS-MLTR (2 GPUs).

**Table 13 sensors-24-01947-t013:** SNR and runtime corresponding to the images displayed in Figure 3.

SNR ^1,2^	Run Time ^4^	HA0	HA100	HA200	HA400	HA800
OS-MLTR_s1 ^3^	6484.32	20.7417	26.3722	30.8319	40.2266	58.2329
OS-MLTR_s5 ^3^	1369.56	10.4133	13.3618	15.4639	20.2634	29.3281
OS-MLTR_s10 ^3^	762.36	8.6491	11.0100	12.8613	16.7671	24.3092

^1^ SNR = *μ*_*ROI*/*σ*_*ROI*. ^2^ Recon. algorithm: OS-MLTR, and All the images were reconstructed with stopping criterion < 0.015. ^3^ Recon. projection: s1 was 600 projections, s5 was 1/5 of s1, s10 was 1/10 of s1. ^4^ Time unit: s.

## Data Availability

Data underlying the results presented in this paper are not publicly available at this time but may be obtained from the authors upon reasonable request.

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
