# Peer review of "Iterative Reconstruction of Micro Computed Tomography Scans Using Multiple Heterogeneous GPUs"

_sensors, 2024, doi:10.3390/s24061947_

Round 1

Reviewer 1 Report (Previous Reviewer 3)

Comments and Suggestions for Authors

The experimental design and results still need significant revisions. Comparing with FDK is unnecessary as it cannot serve as a suitable baseline. Instead, the baseline should be established using OSEM with either multiple homogeneous GPUs or a single GPU. Given that your method does not enhance image quality, analyses related to image quality are not necessary. While Table 12 can support your method, Tables 13 and Figure 3 are irrelevant in this context. In Table 12, it would be beneficial to include a comparison with OSEM using multiple homogeneous GPUs. If this comparison is added, it raises the question of whether there are still advantages for Heterogeneous GPUs compared to homogeneous GPUs.

Author Response

Reviewer 2 Report (Previous Reviewer 1)

Comments and Suggestions for Authors

The problem with the results still persists. The authors evaluate SNR but the reconstructions do not have the same spatial resolution and/or the authors never demonstrate that they have. As explained before, the spatial resolution in case of ART is a function of iteration number. Just because the matrix and voxel size of the FDK and ART volumes are the same does not mean that the spatial resolution is the same. It is evident from figures 2C and 2D that the resolution is vastly different. This, for example, could be evaluated by plotting an edge spread function over the table etc. Also, the window/level stettings are different between 2C and 2D but should be the same since the CT-values should not change. This further complicates things for the readers and an SNR evaluation at different resolution levels ist just not meaningful.

Comments on the Quality of English Language

ok

Round 2

Reviewer 1 Report (Previous Reviewer 3)

Comments and Suggestions for Authors

Again, comparing with FDK is unnecessary as it cannot serve as a suitable baseline. Your paper now is misleading for readers that your algorithm can improve the image quality. Multiple homogeneous GPUs with OSEM is a well established method which should serve as your baseline. 

Author Response

Please see teh attachment

Reviewer 2 Report (Previous Reviewer 1)

Comments and Suggestions for Authors

The authors accounted for all my comments.

Comments on the Quality of English Language

ok

Author Response

Reviewer 2 thoughtyou accounted for all his comments.

Round 3

Reviewer 1 Report (Previous Reviewer 3)

Comments and Suggestions for Authors

I have no more comments. 

This manuscript is a resubmission of an earlier submission. The following is a list of the peer review reports and author responses from that submission.

Round 1

Reviewer 1 Report

Comments and Suggestions for Authors

In this manuscript, the authors try to propose an implementation of an iterative CT reconstruction on a number of heterogeneous GPUs.

-          What is the motivation for that manuscript or the usage of a number of heterogeneous GPUs? In a commercial product, no one would do that. Also, this prohibits the usage of SLI/NVLink.

-          Introduction: The authors state that various CPU-based methods have been proposed. However, none of the methods are designed for a specific architecture. Also, ART and SART are actual algorithms whereas SIR is a group of algorithms.

-          Introduction: The authors list a variety of methods that they say use the afore mentioned algorithms. However, that is just wrong. Where is FBP/FDK using ART/SART/SIR? I also have no idea why the authors list the general conjugated gradient method.

-          Introduction: The authors describe “on-the-fly system matrix” computation as an innovative technique. I am surprised here. No one is actually storing the system matrix, neither in CPU nor GPU implementations. That would not make any sense at all due to its size and everyone just computes that “on-the-fly” without explicitly mentioning it.

-          Introduction: The introduction is missing a discussion of state-of-the-art iterative reconstruction methods. In a commercial setting, no one would actually use a plain OSEM reconstruction. The algorithms provided by manufacturers often only perform a single or only a few iteration(s).

-          Background: Equations 1-3 require more explanations. In particular, the variables mu and b need to be introduced. It is also very confusing to refer to the projections y as images.

-          Background: The pseudo code is very hard to read. Usually, you do not need all the curly braces if you use proper indentation. Is it really crucial to know what variable is a pointer in most cases? What is SumSM?

-          Background: Implementation notes 1: It still seems that the authors store the system matrix or at least parts of it. Why? Also, no one would store a matrix in a 2D array. So the fact that the authors store it in a 1D array is not really noteworthy.

-          Background: What is the stopping criteria? How many iterations are used?

-          Table 7: How where the GPUs chosen?  What was the reason for the given 3 GPUs? Also see my comment above about actual commercial products.

-          Section 3.2: “kVp” should be “kV” in all cases.

-          Figure 2: The plot on the left side is far too small. Images need to be rotated 90° counterclockwise. Please provide window/level settings for all CT images.

-          Figure 2: All reconstructions seem to be misaligned. This is particularly evident in case of the iterative reconstructions.

-          Figure 2: In particular the iterative reconstruction show an increase of CT-values at the bottom of the table and the top of the phantom. Why is that?

-          Figure 2: Was a water precorrection applied to the measured data in either of the cases?

-          Figure 3: Is the stopping criterion and the number of iterations constant for all investigated cases? Where the algorithms fully converged?

-          Table 8: Is the spatial resolution of FDK and OS-MLTR the same? The comparison of SNR would be meaningless otherwise.

-          Figure 4A: That content of this console output does not seem to be discussed in the paper. It shows a GPU utilization between 25% to 37%. So, why use three GPUs at all and not just a single one at 100%? Also the low utilization hints towards a very unoptimized implementation. This also applies to the memory utilization.

-          Figure 4B: What is the motivation for this plot? No one would store the system matrix. This is well known.

-          Discussion: The authors state that the proposed method is bound by the slowest GPU since apparently all faster GPUs have to wait for the slowest one. I wonder what the benefit of the proposed method then is?

-          General: One of the major benefits of iterative methods is the incorporation of physical prior knowledge or a regularization of the image. How, for example, would a TV regularization be implemented over multiple GPUs?

-          General: I feel that a data flow diagram would be a huge improvement to the paper. Also, it seems that all GPUs hold the entire volume in memory. This, of course, is not very efficient and the reason why other methods, e.g. block coordinate descent approaches, are investigated.

-          Implementation: What CUDA version is running on the GPUs? Does it already provide the sophisticated data streaming methods seen in current GPUs?

-          General: The paper does not seem to contain any timing measurements. Why is that? The overall goal seems to be to improve performance.

Comments on the Quality of English Language

Reviewer 2 Report

Comments and Suggestions for Authors

This manuscript developed a finite state automaton (FSA) method that facilitates iterative reconstruction by a heterogeneous multi-GPU platform by parallelizing the matrix calculations derived from a ray tracing system of ordered subsets. Overall, this paper is well written and structured. However, this paper has some shortcomings in regards to text, and the limitations of the current research status of iterative reconstruction methods based on multi-GPU are rarely mentioned in the introduction. In my opinion, the manuscript is suitable for publication after the authors have addressed the comments in file.

Comments on the Quality of English Language

Minor editing of English language required.

Reviewer 3 Report

Comments and Suggestions for Authors

The paper proposed a new method for iterative reconstruction by multiple heterogeneous GPUs. The paper is well organized and the language is fluent and easy to read. My comments are shown below.

Major comments:

1. Why heterogeneous GPUs are necessary? There are other options like multi homogeneous GPUs, a single GPU, GPU/CPU hybrid systems. What are the advantages of heterogeneous GPUs?

2. Could you clearly define the benefits for heterogeneous GPUs? I understand there could be benefits on acceleration and reduced memory. But I got confused on any benefit related to image quality. Could you please 1. confirm whether it has benefits on image quality with the same algorithm (a single GPU) 2. if so, why?

3. In your experiments, why did you decide to compare FDK algorithm with your method (OSEM)? Could you please explain the decision? From my understanding, a fair comparison will be OSEM homogeneous GPUs vs heterogeneous GPUs on the same dose and with the same algorithm. Thera are papers about multi-GPUs and GPU/CPU hybrid OSEM algorithm. I think you need to compare with those.

Minor comments:

1. Abstract: FDK, please provide definition.

2. Abstract: "The proposed method had an SNR six times that of the FDK algorithm at the normal dose and seven times that of the FDK algorithm at the low dose." 

Please add the dose level to the proposed method.

3. Introduction: "The proposed method was developed to mitigate the factors that affect statistical iterative reconstruction, obtain high-quality images from low-dose partial scan views, and reduce the reconstruction time, data transfer overhead, and dependency on a high-capacity DRAM."

Please list the factors that affect statistical iterative reconstruction and can be solved by heterogeneous GPUs. And please explain how heterogeneous GPUs helps "obtain high-quality images from low-dose partial scan views" 

4. 3.2. Experiments: Please provide detailed parameters and implementations for FDK as it is your reference.

5. Discussion and Conclusions:

"In experiments, our proposed method demonstrated a drawback in that the image reconstruction performance with multiple GPUs was lower than with a single GPU (i.e., NVIDIA Titan-XP)."

Where did you compare your method with a single GPU method? Lower performance means speed, memory size or image quality?

Round 2

Reviewer 1 Report

Comments and Suggestions for Authors

The authors revised the paper but only a few of the relevant comments made it to the manuscript.

Major points still include the lack of motivation as well as errors in the evaluation. E.g., the authors state that the iterative method and FDK have the same spatial resolution since they use the same data. This of course is not the case and not how iterative reconstruction methods work. The image quality is a function of iterativen number etc. The geometry used for all reconstructions is stil misaligned. Why not just correct for the misalignment before submission?

Comments on the Quality of English Language

Reviewer 3 Report

Comments and Suggestions for Authors

The main innovation of this paper lies in the utilization of multiple heterogeneous GPUs. However, I still believe that your experimental design has significant issues. 

Your comparison between FDK and OSEM does not support your paper's main novelty, and cannot prove using heterogeneous GPUs is a better method compared to existing ones (e.g. single GPU OSEM).

All your work is unrelated to image quality and only focuses on run-time, speed, memory, and accessibility. The results you present only demonstrate that OSEM is better than FDK, which is not directly aligned with the key point of your paper and has been proved before.
